# Generative Image as Action Models

**Mohit Shridhar**[1,*], **Yat Long Lo**[1,*], **Stephen James**[1]
[1]Dyson Robot Learning Lab, *Equal Contribution

genima-robot.github.io

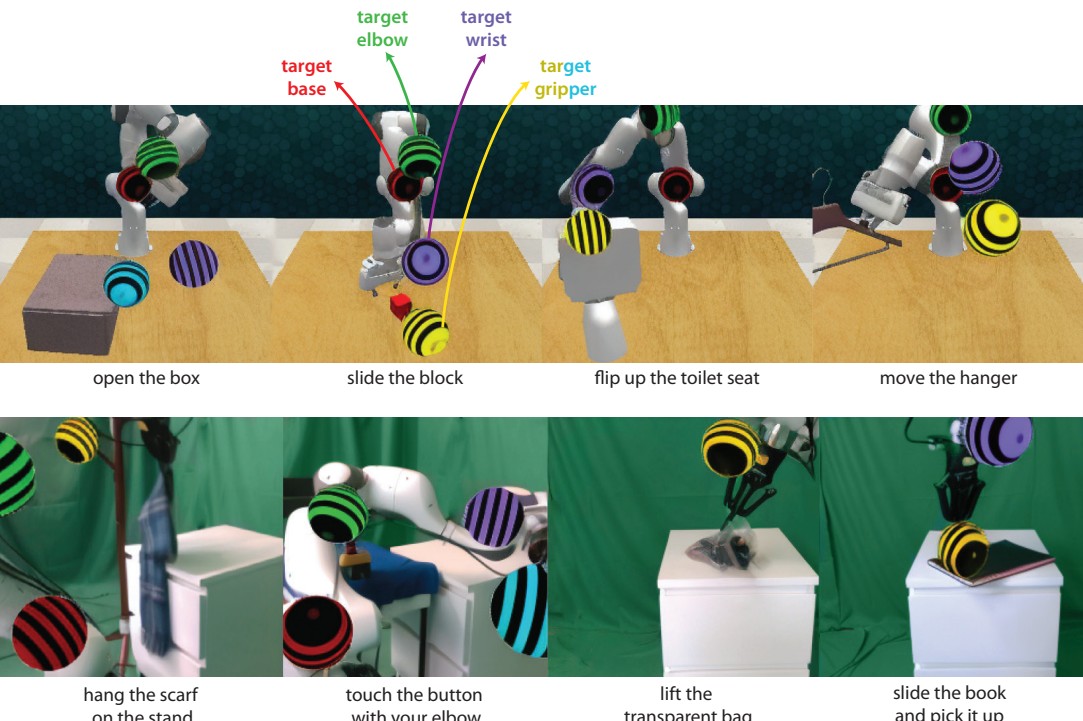

| open the box | slide the block | flip up the toilet seat | move the hanger |

| hang the scarf on the stand | touch the button with your elbow | lift the transparent bag | slide the book and pick it up |

**Abstract:** Image-generation diffusion models have been fine-tuned to *unlock* new capabilities such as image-editing and novel view synthesis. Can we similarly *unlock* image-generation models for visuomotor control? We present GENIMA, a behavior-cloning agent that fine-tunes Stable Diffusion to ***"draw joint-actions"*** as targets on RGB images. These images are fed into a controller that maps the visual targets into a sequence of joint-positions. We study GENIMA on 25 RLBench and 9 real-world manipulation tasks. We find that, by lifting actions into image-space, internet pre-trained diffusion models can generate policies that outperform state-of-the-art visuomotor approaches, especially in robustness to scene perturbations and generalizing to novel objects. Our method is also competitive with 3D agents, despite lacking priors such as depth, keypoints, or motion-planners.

**Keywords:** Diffusion Models, Image Generation, Behavior Cloning, Visuomotor

## 1 Introduction

Image-generation diffusion models [1, 2, 3] are generalists in producing visual-patterns. From photo-realistic images [4] to abstract art [5], diffusion models can generate high-fidelity images by distilling massive datasets of captioned images [6]. Moreover, if both inputs and outputs are in image-space, these models can be fine-tuned to *unlock* new capabilities such as image-editing [7, 8], semantic correspondences [9, 10], or novel view synthesis [11, 12]. Can we similarly *unlock* image-generation models for generating robot actions?

8th Conference on Robot Learning (CoRL 2024), Munich, Germany.

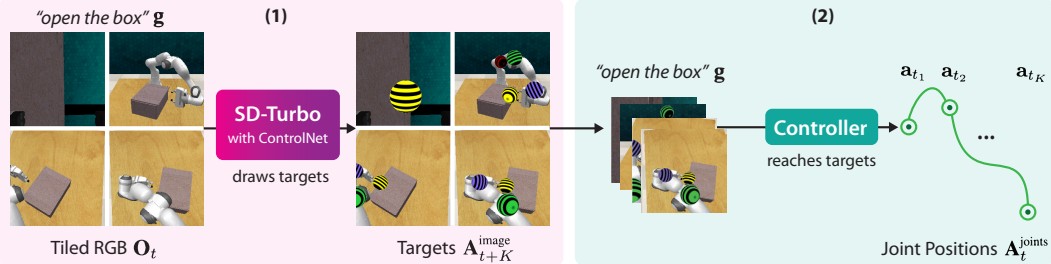

*Figure 1.* **GENIMA Overview.** GENIMA is a behavior-cloning agent that maps multi-view RGB observations and language goals to joint-position actions. GENIMA is composed of two stages: **(1)** SD-Turbo [27] is fine-tuned with ControlNet [8] to ***draw target joint-positions***, which are from the $t + K$ timestep in expert demonstrations. Each joint is rendered as a uniquely colored sphere. **(2)** The generated targets are input into an ACT [19, 23] controller, which translates them into a sequence of $K$ joint-positions. The controller is trained to ignore background context by using random backgrounds (see Figure 2). Both stages are trained independently and used sequentially during inference.

Prior works in robotics have used image-generation for subgoal generation [13, 14, 15, 16], data-augmentation [17, 18, 19], and features-extraction for 3D agents [20, 21]. Subgoal generation predicts goal images as targets, however, producing exact pixel-level details about interactions with deformable objects, granular media, and other chaotic systems, is often infeasible. Data-augmentation methods randomize scenes with image-generation outputs to improve robustness to lighting, textures, and distractors, but they do not use image-generation models to directly generate actions. 3D agents use pre-trained features from diffusion models to improve generalization, however, they rely on privileged information such as depth, keypoints, task-specific scene bounds, and motion-planners.

In this work, we use image-generation models in their native formulation: drawing images. We present GENIMA, a multi-task behavior-cloning agent that directly fine-tunes Stable Diffusion [1] to ***"draw joint-actions"***. To supervise the fine-tuning, we format expert demonstrations into an image-to-image dataset. The input is an RGB image with a language goal, and the output is the same image with joint-position targets from a future timestep in the demonstration. The targets are *rendered* as colored spheres for each joint (as shown in the previous page). These visual targets are fed into a controller that maps them to a sequence of joint-positions. This formulation frames action-generation as an image-generation problem such that action-patterns become visual-patterns.

We study GENIMA on 25 simulated and 9 real-world tasks. In RLBench [22], GENIMA outperforms state-of-the-art visuomotor approaches such as ACT [19, 23] in $16/25$ tasks, and DiffusionPolicies [24] in $25/25$ tasks. More than task performance, we show that GENIMA is robust to scene perturbations like randomized object colors, distractors, and lighting changes, and also in generalizing to novel objects. We find that RGB-to-joint methods can approach the performance of privileged 3D next-best-pose methods [25, 26] without using priors like depth, keypoints, or motion-planners. We validate these results with real-world tasks that involve dynamic motions, full-body control, transparent and deformable objects. In summary, our contributions are:

- A novel problem-formulation that frames joint-action generation as image-generation.
- A proof-of-concept system for drawing and executing joint-actions.
- Empirical results and insights from simulated and real-world experiments.

Our code and pre-trained checkpoints are available at `genima-robot.github.io`.

## 2 GENIMA

GENIMA is a behavior-cloning agent that maps RGB observations $\mathbf{O}_t$ and a language goal $\mathbf{g}$ to joint-position actions $\mathbf{A}_t^{\text{joints}}$. The key objective is to lift actions into image-space such that internet-pretrained diffusion models can learn action-patterns as visual-patterns. This is accomplished through a two-stage process: $(1)$ fine-tuning Stable Diffusion [1] to draw target joint-positions $\mathbf{A}_{t+K}^{\text{image}}$ on input images, $K$ timesteps ahead, and $(2)$ training a controller to translate these targets into to a sequence of executable joint-positions $\mathbf{A}_t^{\text{joints}} = \{\mathbf{a}_{t_1}, \mathbf{a}_{t_2}... \mathbf{a}_{t_K}\}$. This simple two-stage process offloads semantic and task-level reasoning to a generalist image-generation model, while the controller reaches nearby joint-positions indicated by the visual targets. The sections below describe the two stages in detail, and Figure 1 provides an overview.

## 2.1 Diffusion Agent

The diffusion agent controls what to do next. The agent takes language goal $\mathbf{g}$ and multi-view RGB images $\mathbf{O}_t$ as input, and outputs target joint-positions $\mathbf{A}_{t+K}^{\text{image}}$ on the same images. This problem formulation is a classic image-to-image setting, so any fine-tuning pipeline [7, 28, 29] can be used. We specifically use ControlNet [8] to preserve spatial-layouts and for data-efficient fine-tuning.

**Fine-Tuning Data.** To supervise the fine-tuning, we randomly sample observations and target joint-positions from $t + K$ timesteps in expert demonstrations. For that timestep, we obtain 6-DoF poses of each robot joint, which is available through robot-APIs (from forward-kinematics). We place spheres at those poses with `pyrender`[1], and render them on four camera observations $\mathbf{O}_t = \{\mathbf{o}_t^{\text{front}}, \mathbf{o}_t^{\text{wrist}}, \mathbf{o}_t^{\text{left}}, \mathbf{o}_t^{\text{right}}\}$ with known intrinsics and extrinsics. On a 7-DoF Franka Panda, we only render four joints: `base`, `elbow`, `wrist`, and `gripper`, to avoid cluttering the image. Each joint is represented with an identifying color, with separate colors for gripper `open` and `close`. The spheres include horizontal stripes parallel to the joint's rotation axis, acting as graduations indicating the degree of rotation. Only horizontal stripes are necessary as each joint has only one rotation axis. The stripes are also asymmetric across the poles to help identify the sphere orientation.

**Fine-Tuning with ControlNet.** Given an image-to-image dataset, we finetune Stable Diffusion [27] with ControlNet [8] to draw targets on observation images. ControlNet is a two-stream architecture: one stream with a frozen Stable Diffusion UNet that gets noisy input and language descriptions, and a trainable second stream that gets a conditioning image to modulate the output. This architecture retains the text-to-image capabilities of Stable Diffusion, while fine-tuning outputs to spatial layouts in the conditioning image. GENIMA uses RGB observations as the conditioning image to draw precise targets. We use SD-Turbo [27] – a distilled model that can generate high-quality images within 1 to 4 diffusion steps – as the base model for fine-tuning. We use the HuggingFace implementation [30][2] of ControlNet without modifications. See Appendix D for more details on fine-tuning.

**Tiled Diffusion.** Fine-tuning Stable Diffusion on robot data poses three key challenges. Firstly, Stable Diffusion models work best with image resolutions of $512 \times 512$ or higher due to their training data. In robotics, large images increase inference latency. Secondly, multi-view generation suffers from inconsistencies across viewpoints. However, multi-view setups are crucial to avoid occlusions and improve spatial-robustness. Thirdly, diffusion is quite slow, especially for generating four target images at every timestep. Inspired by view-synthesis works [31, 32], we solve all three challenges with a simple solution: tiling. We tile four observations of size $256 \times 256$ into a single image of $512 \times 512$. Tiling generates four images at 5 Hz on an NVIDIA A100 or 4 Hz on an RTX 3090.

## 2.2 Controller

The controller translates target images $\mathbf{A}_{t+K}^{\text{image}}$ into executable joint-positions $\mathbf{A}_t^{\text{joints}}$. The controller can be implemented with any visuomotor policy that maps RGB observations to joint-positions. We specifically use ACT [23, 33] – a Transformer-based policy architecture [34] – for its fast inference-speed and training stability. However, in theory, any framework like Diffusion Policies [24] or RL-based methods [35] can be used. Even classical controllers can be used if pose-estimates of target spheres are provided, but in our implementation, we opt for learned controllers for simplicity.

**Training.** During training, the controller receives current joint-positions, the language goal, and RGB images with ground-truth targets overlaid on random backgrounds. The random backgrounds, as shown in Figure 2, force ACT to follow targets and ignore any contextual information in the scene. We use the same hyperparameters and settings from the original ACT codebase[3] with minor modifications. To improve robustness to fuzzy diffusion outputs, we augment images with random-crops [36], color

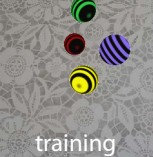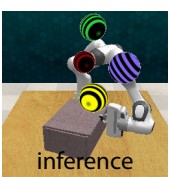

*Figure 2.* During training, the controller's input are ground-truth targets with random backgrounds (left). During inference, the targets are from the diffusion agent (right).

---

[1] https://pyrender.readthedocs.io/en/latest/examples/quickstart.html
[2] https://huggingface.co/docs/diffusers/en/using-diffusers/controlnet
[3] https://github.com/tonyzhaozh/act

jitters, elastic transforms, and Gaussian noise. We use L1 loss for joint-actions, and cross-entropy loss for gripper open and close actions. For language-conditioning, we use FiLM [37] layers following MT-ACT [19]. The controller is trained independently from the diffusion agent. See Appendix F for hyperparameters and Appendix E for controller details.

**Inference.** During inference, the controller gets target images from the diffusion agent. The controller predicts a sequence of $K$ joint-actions, and executes $K$ actions or less before querying the diffusion agent in a closed-loop fashion. The controller runs at $\sim 50$ Hz on an NVIDIA RTX 3090.

## 3 Experiments

We study GENIMA in both simulated and real-world environments. Specifically, we are interested in answering the following questions:

§ 3.1 How does GENIMA compare against state-of-art visuomotor policies and 3D baselines?

§ 3.2 What are the benefits of drawing actions with internet-pretrained image-generation models?

§ 3.3 Which factors affect GENIMA's performance?

§ 3.4 How well does GENIMA perform on real-world tasks?

We start with benchmarking our method in simulated environments for reproducible and fair comparisons. The following sections describe our simulation setup and evaluation methodology.

**Simulation Setup.** All simulated experiments are set in CoppeliaSim [38] interfaced through PyRep [39]. The robot is a 7-DoF Franka Emika Panda placed behind a tabletop. Observations are captured from four RGB cameras: `front`, `left_shoulder`, `right_shoulder`, and `wrist`, each with a resolution of $256 \times 256$. The robot is commanded with joint-position actions via PID control, or end-effector actions via an IK solver.

**25 RLBench Tasks.** We choose 25 (out of 100) tasks from RLBench [22]. While most RLBench tasks are suited for discrete, quasi-static motions that benefit 3D next-best-pose agents [25, 26], we pick tasks that are difficult to execute with end-effector control. Tasks such as `open box` and `open microwave` require smooth non-linear motions that sampling-based motion-planners and IK solvers struggle with. Each RLBench task includes several variations, but we only use `variation0` to reduce training time with limited resources. However, our method should be applicable to multi-variation settings without any modifications. We generate two datasets: 50 training demos and 50 evaluation episodes per task. For both datasets, objects are placed randomly, and each episode is sanity-checked for solvability. Language goals are constructed from instruction templates. See Appendix A for details on individual tasks.

**Evaluation Metric.** Multi-task agents are trained on all 25 tasks, and evaluated individually on each task. Scores are either 0 for failures or 100 for successes, with no partial successes. We report average success rates on 50 evaluation episodes across the last three epoch checkpoints: $50 \times 3 = 150$ episodes per task, which adds up to $150 \times 25 = 3750$ in total. We use a single set of checkpoints for all tasks without any task-specific optimizations or cherry-picking.

**Visuomotor Baselines.** We benchmark GENIMA against three state-of-the-art visuomotor approaches: ACT [19, 23], DiffusionPolicies [24], and SuSIE [13]. ACT is a transformer-based policy that has achieved compelling results in bimanual manipulation. Although GENIMA uses ACT as the controller, our controller has never seen RGB observations from demonstrations, just sphere targets with random backgrounds. DiffusionPolicies is a widely adopted visuomotor approach that generates multi-modal trajectories through diffusion. SuSIE is the closest approach to GENIMA, but instead of drawing target actions, SuSIE generates target RGB observations as goal images. We adapt SuSIE to our setting by training a controller that maps target and current RGB observations to joint-position actions. All multi-task baselines are conditioned with language goals. ACT and DiffusionPolicies use FiLM conditioning [37], whereas SuSIE uses the goal as a prompt.

| Task | GENIMA | ACT | SuSIE | Diff. Policy | 3D Diff. Actor |
|------|--------|-----|-------|--------------|----------------|
| basketball in hoop | **50.0** | 32.7 | 5.3 | 0.0 | 100 |
| insert usb | **26.0** | 18.0 | 0.0 | 0.0 | 29.3 |
| move hanger | **94.0** | 42.0 | 21.3 | 0.0 | 76.0 |
| open box | **79.3** | 69.3 | 36.0 | 3.3 | 6.0 |
| open door | **85.3** | 75.3 | 26.6 | 6.7 | 76.7 |
| open drawer | 77.3 | **82.7** | 67.3 | 0.0 | 71.3 |
| open grill | **48.7** | 40.0 | 26.0 | 0.0 | 93.3 |
| open microwave | **46.7** | 22.6 | 10.0 | 0.0 | 58.0 |
| open washer | **46.0** | 22.6 | 2.0 | 2.0 | 82.7 |
| open window | **69.3** | 8.0 | 24.6 | 0.0 | 96.7 |
| phone on base | **18.7** | 13.3 | 1.0 | 2.0 | 94.0 |
| pick up cup | 36.0 | **43.3** | 24.6 | 0.7 | 92.7 |
| play jenga | 90.0 | **99.3** | 40.0 | 1.3 | 92.0 |
| press switch | 72.7 | 65.3 | **74.7** | 22.7 | 83.3 |
| push button | **76.7** | 31.3 | 7.3 | 2.7 | 46.7 |
| put books on shelf | 14.7 | **44.0** | 1.0 | 0.0 | 36.7 |
| put knife on board | 12.7 | **14.7** | 4.0 | 2.0 | 77.3 |
| put rubbish in bin | **26.7** | 15.3 | 6.7 | 0.0 | 96.0 |
| scoop with spatula | 11.3 | **22.7** | 1.0 | 0.0 | 66.0 |
| slide block | 12.7 | **22.0** | 0.0 | 0.0 | 99.3 |
| take lid off | 48.0 | 44.7 | **72.0** | 1.3 | 100 |
| take plate off | 21.3 | **37.3** | 4.0 | 0.0 | 72.7 |
| toilet seat up | **93.3** | 45.3 | 50.0 | 2.0 | 94.0 |
| turn on lamp | 12.0 | **19.3** | 6.0 | 4.0 | 4.0 |
| turn tap | **71.3** | 59.3 | 32.7 | 20.7 | 99.3 |
| average | **49.6** | 39.6 | 21.8 | 2.9 | 73.8 |

Table 1. **Visuomotor and 3D Baselines on 25 RLBench tasks.** Success rates (%) for multi-task agents trained with 50 demos and evaluated on 50 episodes per task. We report average scores across the last three checkpoints. The four methods on the left are RGB-to-joint agents. The rightmost method is a 3D next-best-pose agent with extra priors: depth, keypoints, scene bounds, and motion-planners.

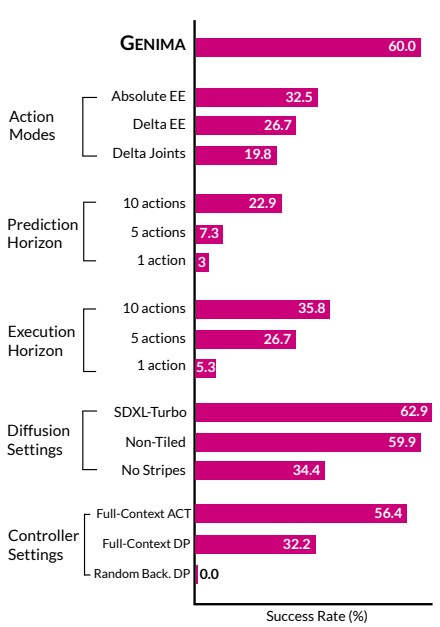

Figure 3. **Ablations and Sensitivity Analyses.** We study factors that affect GENIMA's performance by training a multi-task agent on 3 tasks: `take lid off`, `open box`, and `slide block`. We report average success rates across the 3 tasks.

**3D Baseline.** We also benchmark GENIMA against 3D Diffuser Actor [40] – a state-of-the-art agent in RLBench. 3D Diffuser Actor uses CLIP [41] to extract vision and language features, and diffuses end-effector poses with a 3D transformer. We use 3D Diffuser Actor as the best-performing representative of 3D next-best-pose agents [25, 26] such as PerAct [42], Hiveformer [43], RVT [44], Act3D [45], DNAct [20], and GNFactor [21]. These works rely on several priors: depth cameras, motion-planners to reach poses, keypoints that segment trajectories into bottlenecks, task-specific scene bounds, and quasi-static assumption for motions.

### 3.1 Visuomotor and 3D Baselines

Our key result is that we show GENIMA – an image-generation model fine-tuned to draw actions – *works at all* for visuomotor tasks. In the sections below, we go beyond this initial result and quantify GENIMA's performance against state-of-the-art visuomotor and 3D baselines.

**GENIMA outpeforms ACT, DiffusionPolicies, and SuSIE.** Table 1 presents results from RLBench evaluations. GENIMA outperforms ACT [19, 23] in 16/25 tasks, particularly in tasks with occlusions (e.g., `open window`) and complex motions (e.g., `turn tap`). Against SuSIE [13], GENIMA performs better on 23/25 tasks, as SuSIE struggles to generate exact pixel-level details for goals. DiffusionPolicy [24] performs poorly in multi-task settings with joint-position control. We ensured that our implementation is correct by training DiffusionPolicy on just `take lid off`, which achieved a reasonable success rate of 75%, but we could not scale it to 25 tasks.

**RGB-to-joint agents approach the performance of 3D next-best-pose agents.** Without 3D input and motion-planners, most prior works [42, 46] report zero-performance for RGB-only agents in RLBench. However, our results in Table 1 show that RGB-to-joint agents like GENIMA and ACT can be competitive with 3D next-best-pose agents. GENIMA outperforms 3D Diffuser Actor in 6/25 tasks, particularly in tasks with non-linear trajectories (e.g., `open box`, `open door`), and tiny objects (e.g., `turn on lamp`). GENIMA also performs comparably (within 3%) on 3 more tasks: `insert usb`, `play jenga`, `toilet seat up`, despite lacking priors. 3D Diffuser Actor performs better on most tasks, but training GENIMA for longer or with more data could bridge this gap.

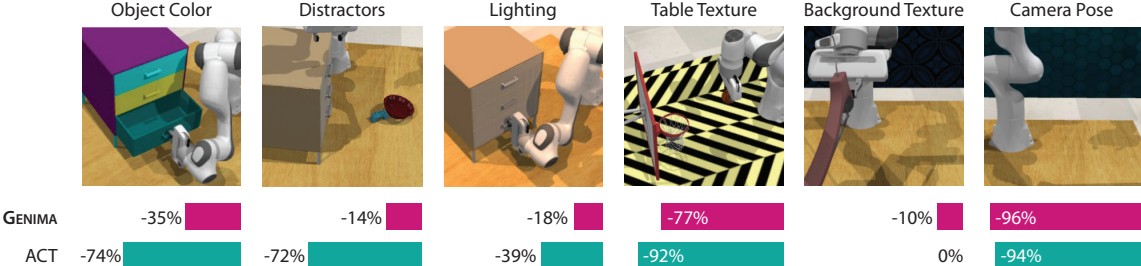

| | Object Color | Distractors | Lighting | Table Texture | Background Texture | Camera Pose |
|---|---|---|---|---|---|---|
| GENIMA | -35% | -14% | -18% | -77% | -10% | -96% |
| ACT | -74% | -72% | -39% | -92% | 0% | -94% |

*Figure 4.* **Performance drops from Colosseum [46] perturbations.** We evaluate GENIMA and ACT on 6 perturbation categories: randomized object and part colors, distractor objects, lighting color and brightness variations, randomized table textures, randomized backgrounds, and camera pose changes. We report success rates from 150 evaluation episodes per task, where perturbations are randomly sampled episodically. ACT overfits to objects and lighting conditions, whereas GENIMA is more robust to such perturbations. See supplementary video for examples.

## 3.2 Semantic and Spatial Generalization

While all evaluations in Section 3.1 train and test on the same environment, the key benefit of using image-generation models is in improving generalization of visuomotor policies. In this section, we examine semantic and spatial generalization aspects of visuomotor policies.

**GENIMA is robust to semantic perturbations on Colosseum tasks.** We evaluate the same multi-task GENIMA and ACT agents (from Section 3.1) on 6 perturbation categories in Colosseum [46]: randomized object and part colors, distractor objects, lighting color and brightness variations, randomized table textures, randomized scene backgrounds, and camera pose changes. Figure 4 presents results from these perturbation tests. Despite being initialized with a pre-trained ResNet [47], ACT overfits and significantly drops in performance with changes to object color, distractors, lighting, and table textures. Whereas GENIMA has minimal drops in performance from an emergent property that reverts scenes to canonical textures and colors from the training data. See supplementary videos for examples. However, both methods fail to generalize to unseen camera poses.

**GENIMA extrapolates to spatial locations with aligned image-action spaces.** By drawing actions on images, GENIMA keeps the image-space and action-space aligned. This alignment has been shown to improve spatial generalization and data-efficiency in prior works [42, 48, 49]. We observe similar benefits in Figure 5, where ACT struggles in the upper-right region with minimal training examples, but GENIMA succeeds in extrapolating to those locations.

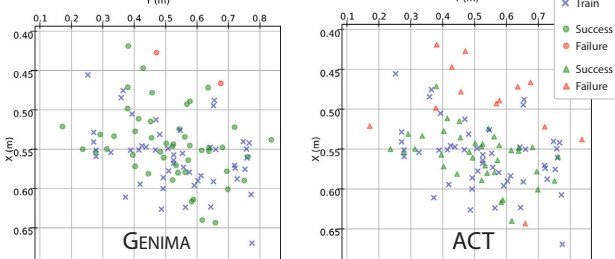

*Figure 5.* **Spatial Generalization.** Train and test saucepan positions (from a top-down view of the tabletop) for evaluations on `take lid off`. ACT struggles to extrapolate to the upper-right region, whereas GENIMA uses aligned image-action spaces for better spatial generalization.

## 3.3 Ablations and Sensitivity Analyses

We investigate factors that affect GENIMA's performance. We report average success rates from multi-task GENIMA trained on 3 tasks: `take lid off`, `open box`, and `slide block`. Our key results are presented in Figure 3, and the sections below summarize our findings.

**Absolute joint-position is the best performing action-mode.** Delta action-modes accumulate errors, and end-effector control through IK struggles with non-linear trajectories. Joint-position actions are also more expressive, allowing for full-body control and other embodiments.

**Longer action sequence predictions are crucial.** In line with prior works [23, 24], modeling trajectory distributions requires predicting longer action sequences. We find that predicting $K = 20$ actions is optimal, since observations are recorded at 20Hz in RLBench.

**Longer execution horizons avoid error accumulation.** Shorter execution horizons lead to jerky motions that put the robot in unfamiliar states. We find that executing all 20 actions works best.

**SDXL improves performance over SD.** Larger base models such as SDXL-Turbo [27] have more capacity to model action-patterns. Newer Transformer-based models [50] might scale even better.

**Tiled diffusion improves generation speed while keeping performance.** Tiled generation of four target images takes 0.2 seconds, whereas generating individual images takes 0.56 seconds. Both methods achieve similar performance, however tiled generation is more multi-view consistent across a wider set of tasks. See Appendix G for reference.

**Without stripes on spheres, performance drops in half.** These stripes act as graduation indicators for joint-angles. Including stripes helps Stable Diffusion learn joint rotations as a visual-pattern.

**Full-context controllers overfit to observations.** Instead of random backgrounds, if controllers are trained with target spheres overlaid on RGB observations from demos, they tend to ignore the targets, and just use observations for action prediction. This hurts performance and generalization.

**ACT works better than DiffusionPolicy as the controller.** Similar to results in Section 3.1, we find that ACT is better at joint-action prediction than DiffusionPolicy (DP). ACT also has a faster inference speed of 0.02 seconds, whereas DiffusionPolicy takes 0.1 seconds (for 20 diffusion steps).

**GENIMA is data-efficient.** We study data-efficiency by constraining poses of objects in training demos. Following R&D [49], we sample poses in a grid-style that maximizes workspace coverage based on the number of demos. GENIMA achieves $80\%$ of the peak performance with 25 demos.

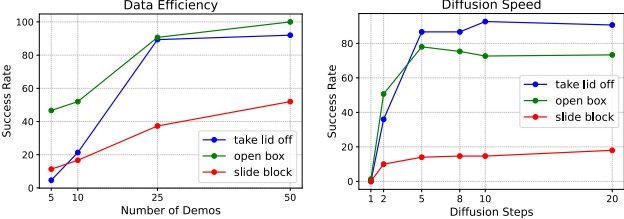

Figure 6. **Data-Efficiency and Diffusion Speed of GENIMA.**

**GENIMA works with 5 diffusion steps or more.** With SD-Turbo [27] as the base model, GENIMA can generate target images with just 5 diffusion steps within 0.2 seconds. Future works can use better schedulers and distillation methods to further improve generation speed and quality.

## 3.4 Real-robot Evaluations

We validate our results by benchmarking GENIMA and ACT on a real-robot setup. Our setup consists of a Franka Emika Panda with 2 external and 2 wrist cameras. We train multi-task agents from scratch on 9 tasks with 50 demos per task. These tasks involve dynamic behaviors (e.g., `slide book`), transparent objects (e.g., `lift bag`), deformable objects (e.g., `hang scarf`), and full-body control (e.g., `elbow touch`). See Figure 7 and supplementary videos for examples. Appendix B covers task details. When comparing GENIMA and ACT, we ensure that the initial state is exactly the same by using an image-overlay tool to position objects. Table 2 reports average success rates from 5 evaluation episodes per task. We also report out-of-distribution performance with unseen objects and scene perturbations. In line with Section 3.2, GENIMA is better than ACT at generalizing to out-of-distribution tasks. GENIMA also exhibits some recovery behavior from mistakes, but DAgger-style [51] training might improve robustness.

| | In-Distribution | | Out-of-Distribution | | |
|---|---|---|---|---|---|
| **Task** | GENIMA | ACT | GENIMA | ACT | *Category* |
| `lid off` | **80** | 60 | **60** | 20 | new saucepan |
| `place teddy` | **100** | 80 | **100** | 60 | new toy |
| `elbow touch` | **80** | 40 | **60** | 40 | new background |
| `hang scarf` | 40 | **60** | **60** | 20 | new scarf |
| `put marker` | **40** | **40** | **40** | 20 | moving objects |
| `slide book` | **100** | 20 | **100** | 0 | darker lighting |
| `avoid lamp` | **40** | 0 | 0 | 0 | new object |
| `lift bag` | **80** | 60 | **40** | **40** | distractors |
| `flip cup` | 20 | **80** | 20 | **40** | new cup |

Table 2. **Real-robot Results.** Success rates of multi-task GENIMA and ACT on 9 real-world tasks, evaluated on 5 episodes per task.

## 4 Related Work

**Visuomotor agents** map images to actions in an end-to-end manner [52, 53, 54]. ACT [23] uses a transformer-based policy architecture [34] to encode ResNet [47] features and predict action-chunks. MT-ACT [19] extends ACT to the multi-task settings with language-conditioning. MVP [55] uses self-supervised visual pre-training on in-the-wild videos and fine-tunes for real-world robotic tasks. Diffusion Policy [24] uses the diffusion process to learn multi-modal trajectories. RT-2 [56] fine-tune vision-language models to predict tokenized actions. Octo [57] can be adapted to new sensory

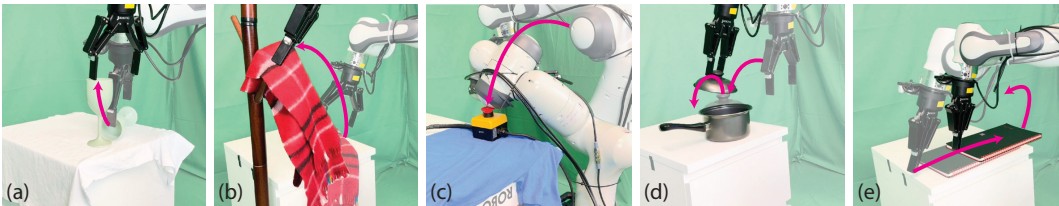

*Figure 7.* **Real-world Tasks.** 5 out of 9 tasks: (a) `flip cup`, (b) `hang scarf`, (c) `elbow touch`, (d) `lid off`, and (e) `slide book`.

inputs and action spaces by mapping inputs into a common tokenized format. All these methods adapt paradigms from vision and language like tokenization and pre-training for predicting actions. In contrast, GENIMA lifts actions back into image-space to use image-generation models natively.

**3D next-best-pose agents** encode 3D input and output 6-DoF poses [25, 26]. These poses are keypoints (or bottlenecks in the trajectory) that are executed with a motion-planner or IK-solver. C2F-ARM [26] and PerAct [42] use calibrated multi-camera setups to voxelize scenes into 3D grids. Voxel grids are computationally expensive, so Act3D [45] and 3D Diffuser Actor [40] replace voxels with sampled 3D points. RVT [44] renders RGB-D input into orthographic projections, and detects actions in image-space. GNFactor [21] and DNAct [20] lift Stable Diffusion features into 3D to improve generalization. Chained Diffuser [58] and HDP [59] use diffusion-based policies as low-level controllers to reach keypoints predicted by 3D next-best-pose agents. All these 3D next-best-pose agents rely on several priors: depth cameras, keypoints, task-specific scene bounds, and/or motion-planners. Whereas GENIMA is a simple RGB-to-joint agent without any of these priors.

**Diffusion models for robot learning.** In addition to modeling policies [60, 61, 62, 63, 64] with diffusion [65, 66], diffusion models have been used in robot learning in various ways. This includes offline reinforcement learning [61, 63, 67], imitation-learning [68, 69, 70], subgoal generation [16, 71, 72], and planning [58, 59, 73, 74, 75, 76]. Others include affordance prediction [77], skill acquisition [78, 79] and chaining [80], reward functions [81], grasping [82], and sim-to-real [83].

**Image-generation models in robot learning** have been used for out-of-distribution data generation [69], and video-conditioned policies [15, 84, 85]. ROSIE [18] and GenAug [17] use image-generation outputs to augment datasets. SuSIE [13], RT-Sketch [86], and RT-Trajectory [87], condition policies on goals generated by image generation models in the form of observations, observation sketches, and trajectory sketches, respectively. GENIMA does not use image-generation models to generate observations, videos, or trajectories, but instead draws joint-actions as targets.

**Representing actions in images** has been shown to improve spatial-robustness and generalization. PIVOT [88] annotates observations with markers to query vision-language models for end-effector actions. RoboTap [89], ATM [90], and Track2Act [91] track dense points on images to learn end-effector policies. R&D [49] renders grippers on images to aid the diffusion process. Yang et al. [92] plan by in-painting navigation actions. C3DM [93] iteratively zooms-into images to predict 6-DoF poses. In comparison, GENIMA does not track points, and learns joint-actions in image-space.

## 5 Conclusion and Limitations

We presented GENIMA, a multi-task agent that fine-tunes Stable Diffusion to draw joint-actions. Our experiments both in simulation and real-world tasks indicate that fine-tuned image-generation models are effective in visuomotor control. While this paper is a proof-of-concept, GENIMA could be adapted to other embodiments, and also to draw physical attributes like forces and accelerations.

GENIMA is quite capable, but not without limitations. Like all BC-agents, GENIMA only distills expert behaviors and does not discover new behaviors. GENIMA also uses camera calibration to render targets, assuming the robot is always visible from some viewpoint. We discuss these limitations and offer potential solutions in Appendix J. But overall, we are excited about the potential of pre-trained diffusion models in revolutionizing robotics, akin to how they revolutionized image-generation.

**Acknowledgments**

Big thanks to the members of the Dyson Robot Learning Lab for discussions and infrastructure help: Abdi Abdinur, Nic Backshall, Nikita Chernyadev, Iain Haughton, Yunfan Lu, Xiao Ma, Sumit Patidar, Younggyo Seo, Sridhar Sola, Eugene Teoh, Jafar Uruç, and Vitalis Vosylius. Special thanks to Tony Z. Zhao and Cheng Chi for open-sourcing ACT and Diffusion Policies, and the HuggingFace Team for `diffusers`.

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

# A    RLBench Tasks

We select 25 out of 100 tasks from RLBench [22] for our simulation experiments. Only `variation0` is used to reduce training time with limited resources. In the following sections, we describe each of the 25 tasks in detail, including any modifications from the original codebase.

## A.1    Basketball in Hoop

**Task:** Pick up the basketball and dunk it in the hoop.
**filename:** `basketball_in_hoop.py`
**Modified:** No.
**Success Metric:** The basketball goes through the hoop.

## A.2    Insert USB in Computer

**Task:** Pick up the USB and insert it into the computer.
**filename:** `insert_usb_in_computer.py`
**Modified:** No.
**Success Metric:** The USB tip is inserted into the USB port on the computer.

## A.3    Move Hanger

**Task:** Move the hanger from one rack to another.
**filename:** `move_hanger.py`
**Modified:** No.
**Success Metric:** The hanger is hung on the other rack and the gripper is not grasping anything.

## A.4    Open Box

**Task:** Grasp the lid and open the box.
**filename:** `open_box.py`
**Modified:** For the data efficiency experiment in Figure 6, we perform grid sampling of box poses for both training and evaluation following R&D [49]. A grid size of 5cm × 20cm is used, with a yaw rotation range of 45° around the $z$ axis. All other experiments use the default random sampling.
**Success Metric:** The joint between the lid and the box is at 90°.

## A.5    Open Door

**Task:** Grip the handle and push the door open.
**filename:** `open_door.py`
**Modified:** No.
**Success Metric:** The door is opened with the door joint at 25°.

## A.6    Open Drawer

**Task:** Open the bottom drawer.
**filename:** `open_drawer.py`
**Modified:** No.
**Success Metric:** The prismatic joint of the button drawer is fully extended.

## A.7    Open Grill

**Task:** Grasp the handle and raise the lid to open the grill.
**filename:** `open_grill.py`
**Modified:** No.
**Success Metric:** The lid joint of the grill cover reaches 50°.

### A.8 Open Microwave

**Task:** Pull open the microwave door.
**filename:** `open_microwave.py`
**Modified:** No.
**Success Metric:** The microwave door is open with its joint reaches 80°.

### A.9 Open Washer

**Task:** Pull open the washing machine door.
**filename:** `open_washing_machine.py`
**Modified:** No.
**Success Metric:** The washing machine door is open with its joint reaches 40 °.

### A.10 Open Window

**Task:** Rotate the handle to unlock the left window, then open it.
**filename:** `open_window.py`
**Modified:** No.
**Success Metric:** The window is open with its joint reaches 30°.

### A.11 Phone On Base

**Task:** Put the phone on the holder base
**filename:** `phone_on_base.py`
**Modified:** No.
**Success Metric:** The phone is placed on the base and the gripper is not holding the phone.

### A.12 Pick Up Cup

**Task:** Pick up the red cup.
**filename:** `pick_up_cup.py`
**Modified:** No.
**Success Metric:** The red cup is picked up by the gripper and held within the success region.

### A.13 Play Jenga

**Task:** Take the protruding block out of the Jenga tower without the tower toppling.
**filename:** `play_jenga.py`
**Modified:** No.
**Success Metric:** The protruding block is no longer on the Jenga tower, the rest of the tower remains standing.

### A.14 Press Switch

**Task:** Flick the switch.
**filename:** `press_switch.py`
**Modified:** No.
**Success Metric:** The switch is turned on.

### A.15 Push Button

**Task:** Push down the maroon button.
**filename:** `push_button.py`
**Modified:** No.
**Success Metric:** The maroon button is pushed down.

### A.16 Put Books on Shelf

**Task:** Pick up books and place them on the top shelf.
**filename:** `put_books_on_bookshelf.py`
**Modified:** No.
**Success Metric:** The books are on the top shelf.

### A.17 Put Knife on Board

**Task:** Pick up the knife and put it on the chopping board.
**filename:** `put_knife_on_chopping_board.py`
**Modified:** No.
**Success Metric:** The knife is on the chopping board and the gripper is not holding it.

### A.18 Put Rubbish in Bin

**Task:** Pick up the rubbish and place it in the bin.
**filename:** `put_rubbish_in_bin.py`
**Modified:** No.
**Success Metric:** The rubbish is inside the bin.

### A.19 Scoop with Spatula

**Task:** Scoop up the block and lift it up with the spatula.
**filename:** `scoop_with_spatula.py`
**Modified:** No.
**Success Metric:** The cube is within the success region, lifted up by the spatula.

### A.20 Slide Block to Target

**Task:** Slide the block towards the green square target.
**filename:** `slide_block_to_target.py`
**Modified:** For the data efficiency experiment in Figure 6, we perform grid sampling of block poses for both training and evaluation following R&D [49]. A grid size of 15cm × 40cm is used, with a yaw rotation range of 90° around the $z$ axis. All other experiments use the default random sampling.
**Success Metric:** The block is in side the green target area.

### A.21 Take Lid off Saucepan

**Task:** Take the lid off the saucepan
**filename:** `take_lid_off_saucepan.py`
**Modified:** For the data efficiency experiment in Figure 6, we perform grid sampling of saucepan poses for both training and evaluation following R&D [49]. A grid size of 35cm × 44cm is used, with a yaw rotation range of 90° around the $z$ axis. All other experiments use the default random sampling.
**Success Metric:** The lid is lifted off from the saucepan to the success region above it.

### A.22 Take Plate off Colored Dish Rack

**Task:** Take the plate off the black dish rack and leave it on the tabletop.
**filename:** `take_plate_off_colored_dish_rack.py`
**Modified:** No.
**Success Metric:** The plate is lifted off the black disk rack and placed within the success region on the tabletop.

### A.23 Toilet Seat Up

**Task:** Lift the lid of the toilet seat to an upright position.
**filename:** `toilet_seat_up.py`
**Modified:** No.
**Success Metric:** The lid joint is at 90°.

### A.24 Turn on Lamp

**Task:** Press the button to turn on the lamp.
**filename:** `lamp_on.py`
**Modified:** No.
**Success Metric:** The lamp is turned on by pressing the button.

### A.25 Turn Tap

**Task:** Grasp the left tap and turn it.
**filename:** `turn_tap.py`
**Modified:** No.
**Success Metric:** The left tap is rotated by 90° from the initial position.

## B Real-World Tasks

We evaluate on 9 real-world tasks. In the following sections, we describe each of 9 tasks in detail, including tests we perform to assess out-of-distribution generalization. Figure 8 shows objects and scene perturbations.

### B.1 Lid Off

**Task:** Take the lid off the saucepan.
**In-Distribution:** A black saucepan with an oval-shaped lid handle seen during training.
**Out-of-Distribution:** An unnseen smaller saucepan with a round-shaped lid handle.
**Success Metric:** The lid is picked up from the saucepan and placed on the right side.

### B.2 Place Teddy

**Task:** Place the teddy into the drawer
**In-Distribution:** A beige-color teddy bear toy seen during training.
**Out-of-Distribution:** An unseen blue plush toy.
**Success Metric:** The toy is inside the drawer.

### B.3 Elbow Touch

**Task:** Touch the red button with the elbow joint.
**In-Distribution:** The button is placed over a blue cloth seen during training.
**Out-of-Distribution:** The button is placed over an unseen pink cloth.
**Success Metric:** The robot touches the button with its elbow joint.

### B.4 Hang Scarf

**Task:** Hang the scarf on the hanger.
**In-Distribution:** A seen green-and-black checkered scarf seen during training.
**Out-of-Distribution:** An unseen red checkered scarf with a different thickness.
**Success Metric:** The scarf hangs still on the lowest peg of the hanger.

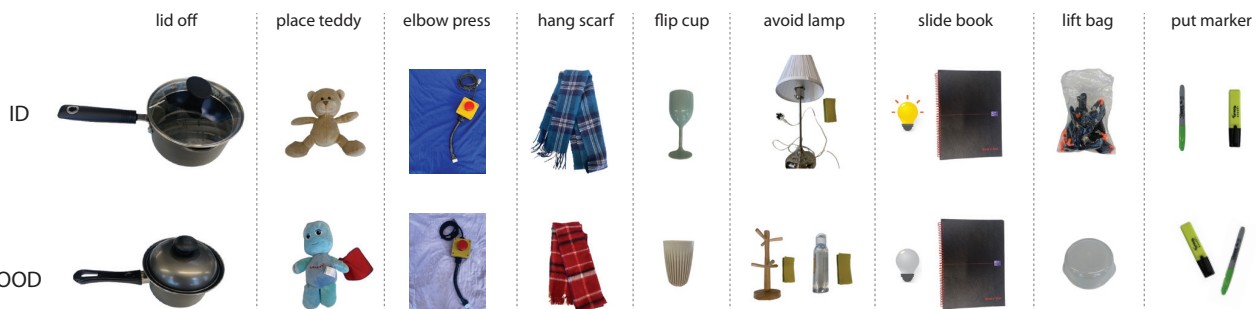

Figure 8. **Real task objects.**. Photos of objects from in-distribution (ID) and out-of-distribution (OOD) evaluations.

## B.5 Put Marker

**Task:** Put the highlighter into the mug.
**In-Distribution:** A highlighter and mug seen during training.
**Out-of-Distribution:** The same highlighter and mug is moved around during execution.
**Success Metric:** The highlighter is inside the mug.

## B.6 Slide Book

**Task:** Slide the book to the drawer's edge and pick it up from the side.
**In-Distribution:** A book seen during training.
**Out-of-Distribution:** The same book and scene but with darker lighting conditions.
**Success Metric:** The book is lifted up from the drawer.

## B.7 Avoid Lamp

**Task:** Pick up the sponge and place it inside the drawer without bumping into the obstacle.
**In-Distribution:** A sponge and lamp (as the obstacle) seen during training.
**Out-of-Distribution:** The same sponge, but with either cup stand or water bottle as the obstacle.
**Success Metric:** The sponge is placed into the drawer without bumping into the obstacle.

## B.8 Lift Bag

**Task:** Lift up the plastic bag.
**In-Distribution:** A plastic bag seen during training.
**Out-of-Distribution:** The same plastic bag, but placed on distractors of different heights.
**Success Metric:** The bag is lifted up from the drawer.

## B.9 Flip Cup

**Task:** Pick up the cup, rotate it, and place it in an upright position.
**In-Distribution:** A plastic wine glass seen during training.
**Out-of-Distribution:** A unseen ceramic coffee cup.
**Success Metric:** The cup is standing upright on the drawer.

## C  Hardware Setup

### C.1  Simulation

Our simulated experiments use a four-camera setup: `front`, `left shoulder`, `right shoulder`, and `wrist`. All cameras are set to default camera poses from RLBench [22] without any modifications, except for the perturbation tests in Section 3.2.

### C.2  Real-Robot

**Hardware Setup.** Real-robot experiments use a 7-DoF Franka Emika Panda equipped with a Robotiq 2F-140 gripper. We use four RealSense D415 cameras to capture RGB images. Two cameras on the end-effector (`upper wrist`, `lower wrist`) to provide a wide field-of-view, and two external cameras (`front`, `right shoulder`) that are fixed on the base. We use a TARION camera mount[4] for the `right shoulder` camera. The extrinsics between the cameras and robot base-frame are calibrated with the `easy handeye` package[5] in ROS.

**Data Collection.** We collect demonstrations for real-world tasks using a joint-mirroring setup similar to ALOHA [23]. Figure 9 shows the data collection setup. A Leader Franka is moved by the operator and the Follower Franka mirrors the Leader's movement in joint space. Visual observations and joint states are recorded at 30 FPS. When training controllers, we set the action prediction horizon to match the data recording frequency to avoid big jumps or slow trajectory execution.

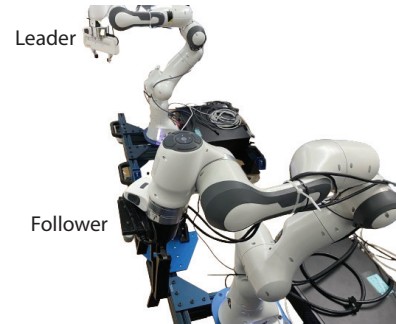

*Figure 9.* Joint-mirroring setup used for data-collection.

### C.3  Training and Evaluation Hardware

The diffusion agents of GENIMA and SuSIE [13], and the 3D Diffuser Actor [40] baseline were trained on a single NVIDIA A100 GPU with 80GB VRAM. The controllers were trained on a single NVIDIA L4 GPU with 24GB VRAM. Evaluation inference for real-world agents was done on an NVIDIA GeForce RTX 3090 GPU with 24GB VRAM.

## D  ControlNet Overview

ControlNet [8] is a fine-tuning architecture that preserves the text-to-image capabilities of Stable Diffusion while following the spatial layout of a conditioning image. ControlNet has achieved compelling results in several image-to-image domains such as sketch-to-image, normal-map-to-image, depth-to-image, canny-edge-to-image, segmentations-to-image, and human-pose-to-image. Particularly, the method preserves spatial structures and can be trained on small datasets. This is achieved through a two-stream architecture similar to prior works like CLIPort [94].

**Two-stream architecture.** ControlNet's architecture is composed of two streams: frozen and trainable. The frozen-stream is a pre-trained copy of Stable Diffusion's UNet, whose parameters are kept frozen throughout fine-tuning. The trainable-stream is another copy of the UNet's downsampling encoder, whose parameters are fine-tuned. The frozen-stream gets sampled latents, a prompt, and time embeddings as input. The trainable-stream gets latents of the conditioning image (that is encoded with a frozen autoencoder), a prompt, and time embeddings as input. The two streams are connected through zero-convolution layers where outputs from each layer of the trainable-stream are added to the decoder layers of the frozen-stream.

---

[4]https://amzn.eu/d/7xDDfJH
[5]https://github.com/IFL-CAMP/easy_handeye

**Zero-convolution connections** regulate the flow of information from the trainable-stream to the frozen-stream. Zero-convolution layers are $1 \times 1$ convs initialized with zeroed weights and biases. At the start of the fine-tuning process, the trainable-stream makes no contribution to the final output because of the zero-intialization. But after fine-tuning, the trainable layers modulate the output to follow the spatial layout in the conditioning image.

**Training Loss.** ControlNet is trained with a standard diffusion loss that predicts noise added to a noise image. This is implemented as an L2-loss on the latents. For more details on the training process, refer to the original ControlNet paper [8].

# E ACT Overview

Action Chunking with Transformers (ACT) [23] predicts action chunks (or sequences) to reduce the effective horizon of long-horizon tasks. This helps alleviate compounding errors in behavior-cloning when learning from human demonstrations. The chunk size is fixed at length $K$. Given an observation, the model outputs the next $K$ actions to be executed sequentially.

**Architecture.** Images are encoded with a pre-trained ResNet-18 [47]. The vision features from each camera and proprioceptive features are then fed into a conditional variational autoencoder (CVAE). The CVAE consists of a BERT-like [95] transformer encoder and decoder. The encoder takes in the current joint position and target action sequence to predict the mean and variance of a style variable $z$. This style variable $z$ helps in dealing with multi-modal demonstrations. It is only used to condition the action decoder during training and is discarded at test time by zeroing it out. The action decoder is based on DETR [34], and is trained to maximize the log-likelihood of action chunks from human demonstrations using two losses: an action reconstruction loss and a KL regularization term to encourage a Gaussian prior for $z$.

**Temporal Smoothing.** To avoid jerky robot motions, ACT [23] uses temporal ensembling at each timestep. An exponential weighted scheme $w_i = exp(-m * i)$ is applied to obtain a weighted average of actions from overlapping predictions across timesteps, where $w_0$ is the coefficient for the oldest action and $m$ controls the speed for incorporating new observations.

**Our modifications to ACT.** We made several modifications to ACT [23] in our implementation to improve data-efficiency and robustness:

- Data augmentation: we use random-crops [36], color jitters, elastic transforms and Gaussian noise from Torchvision[6]. The original ACT [23] does not use any data augmentation.

- Sliding window sampling: we apply a sliding window along each demonstration trajectory to obtain action chunks. The original ACT [23] samples action chunks sparsely from each trajectory. The sliding-window ensures full data-coverage every epoch.

- Discrete gripper loss: we use cross entropy loss for gripper open and close actions instead of an L1-loss. This makes the prediction closer to how the data was collected.

- Temporal ensemble smoothing: we do not use temporal smoothing for our ACT [23] controllers. It oversmoothens trajectories, which reduces precision and recovery behaviors.

- Transformer decoder features: the original implementation conditions action predictions on only the first decoder layer, leaving some unused layers[7]. We replace it with the last decoder feature instead.

---

[6]https://pytorch.org/vision/0.15/transforms.html
[7]https://github.com/tonyzhaozh/act/issues/25

# F  Hyperparameters

In this section, we provide training and evaluation hyperparameters for GENIMA and other baselines. Note that GENIMA's controller, SuSIE [96], and the ACT [23] baseline all share the same hyperparameters and augmentation settings for fair one-to-one comparisons. Real-world GENIMA and ACT agents also use the same hyperparameters (except for the camera setup).

| | |
|---|---|
| Base Model | SD-Turbo [27] |
| Target $K$ timestep | 20 |
| Learning rate | $1e^{-5}$ |
| Weight decay | $1e^{-2}$ |
| Epochs | 200 |
| Batch size | 24 |
| Image size | $512 \times 512$ (tiled) |
| Image augmentation | `color jitter`, `random crop` |
| Train scheduler | DDPM [66] |
| Test scheduler | Euler Ancestral Discrete [97] |
| Learning rate scheduler | constant |
| Learning rate warm-up steps | 500 |
| Inference diffusion steps | 10 (for RLBench) |
| Joints with rendered spheres | `base`, `elbow`, `wrist`, `gripper` |
| Sphere radius for each camera (wrist, front, right, left) | 3cm, 8cm, 6.5cm, 6.5cm |

*Table 3.* Diffusion Agent hyperparameters for GENIMA and SuSIE [13].

| | |
|---|---|
| Backbone | ImageNet-trained ResNet18 [47] |
| Action dimension | 8 (7 joints + 1 gripper open) |
| Cameras | wrist, front, right shoulder, left shoulder |
| Learning rate | $1e^{-5}$ |
| Weight decay | $1e^{-4}$ |
| Image size | $256 \times 256$ |
| Action sequence $K$ | 20 |
| Execution horizon | 20 |
| Observation horizon | 1 |
| # encoder layers | 4 |
| # decoder layers | 6 |
| # heads | 8 |
| Feedforward dimension | 2048 |
| Hidden dimension | 256 |
| Dropout | 0.1 |
| Epochs | 1000 |
| Batch size | 96 |
| Temporal ensembling | False |
| Action Normalization | zero mean, unit variance |
| Image augmentation | `color jitter`, `random crop`, `elastic`, and `Gaussian noise` |

*Table 4.* Controller hyperparameters for GENIMA, SuSIE [13], and ACT [23] baseline.

| | |
|---|---|
| Backbone | ImageNet-trained ResNet18 [47] |
| Noise Predictor | UNet [98] |
| Action Dimension | 8 (7 joints + 1 gripper open) |
| Cameras | wrist, front, right shoulder, left shoulder |
| Learning rate | $1e^{-4}$ |
| Weight decay | $1e^{-4}$ |
| Image size | $256 \times 256$ |
| Observation horizon | 1 |
| Action sequence $K$ | 16 |
| Execution horizon | 16 |
| Train, test diffusion steps | 50, 50 |
| Hidden dimension | 512 |
| Epochs | 1000 |
| Batch size | 128 |
| Scheduler | DDPM [66] |
| Action Normalization | [-1, 1] |
| Image augmentation | `color jitter`, `random crop`, `elastic`, and `Gaussian noise` |

Table 5. Diffusion Policy [24] hyperparameters.

| | |
|---|---|
| Learning rate | $1e^{-4}$ |
| Weight decay | $5e^{-4}$ |
| Action history length | 3 |
| Train, test diffusion steps | 100 |
| Embedding dimension | 120 |
| Training iterations | 550000 |
| Batch size | 8 |
| Position scheduler | scaled linear |
| Rotation scheduler | squared cosine noise |
| Loss weight $w_1$ | 30 |
| Loss weight $w_2$ | 10 |

Table 6. 3D Diffuser Actor [40] hyperparameters.

## G    Tiled vs. Non-Tiled Generation

Figure 10 shows an example of tiled vs non-tiled generation. Non-tiled generation only gets one camera-view input at a time during diffusion. Without the full scene context, non-tiled generation tends to produce inconsistent and ambiguous predictions like duplicate elbow targets.

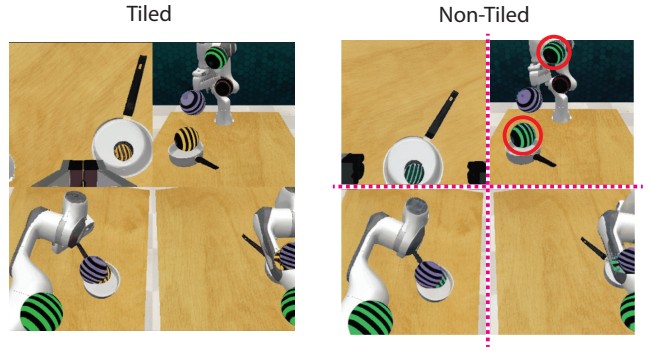

Figure 10. Non-tiled generation makes inconsistent predictions like the duplicate elbow targets highlighted with red circles on the right.

# H    Base Diffusion Models and Fine-Tuning Pipelines

GENIMA's formulation is agnostic to the choice of base Stable Diffusion model and also the fine-tuning pipeline. The SD-Turbo [27] base model used in GENIMA can be replaced with a bigger base model like SDXL-Turbo [27, 5] that is trained on larger images. Likewise, instead of fine-tuning with ControlNet [8], we can also use Instruct-pix2pix [7]. See Figure 11 for examples.

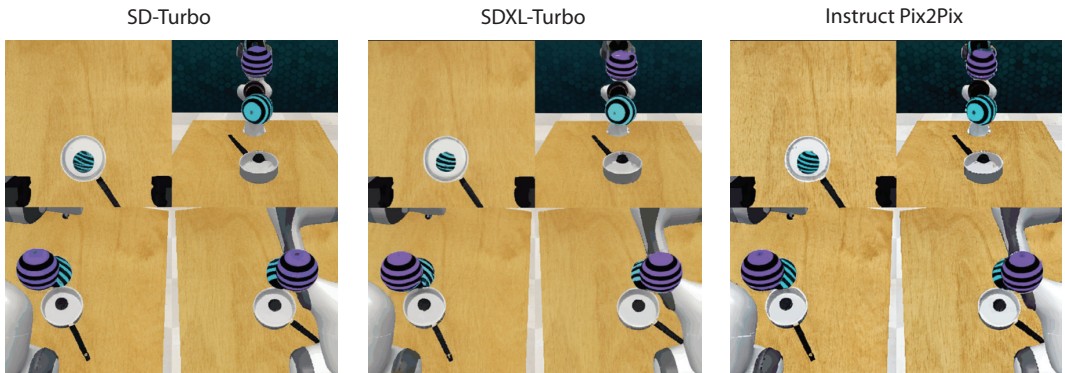

*Figure 11.* Drawing joint-actions with SD-Turbo [27], SDXL-Turbo [27, 5], and Instruct-pix2pix [7].

# I    SuSIE Goal Predictions

Figure 12 shows examples of goal images generated by SuSIE [13] with fined-tuned ControlNet [8]. In general, SuSIE struggles to precisely predict pixel-level details of dynamic scenes with complex object interactions such as `turn tap` and `take plate off`.

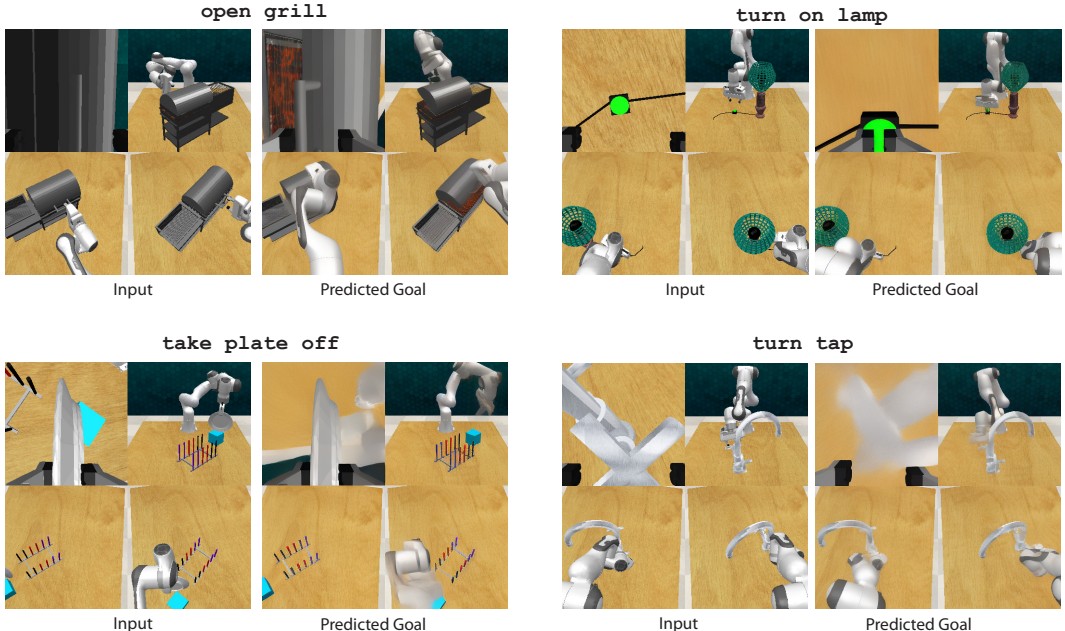

*Figure 12.* Examples of goals predicted by SuSIE [13].

## J   Limitations and Potential Solutions

While GENIMA is quite capable, it is not without limitations. In the following sections, we discuss some of these limitations and potential solutions.

**Camera extrinsics during training.** To create a fine-tuning dataset, GENIMA relies on calibrated cameras with known extrinsics to render target spheres. While the calibration process is quick, it can be difficult to obtain extrinsics for pre-existing datasets or in-the-wild data. A simple solution could be to use camera pose-estimation methods like DREAM [99]. DREAM takes a single RGB image of a known robot and outputs extrinsics with comparable error rates to traditional hand-eye calibration.

**Robot visibility in observations.** One strong assumption our method makes is that the robot is always visible from some viewpoint in order to draw actions near joints. This assumption might not always hold, especially in camera setups with heavy occlusion or wrist-only input. A potential solution could be to provide a virtual rendering of the robot-state, which is commonly available from visualization and debugging tools like RViz[8]. The virtual rendering can be tiled with observations such that the diffusion agent can incorporate both the virtual robot-state and observations. See Figure 13 for an illustration.

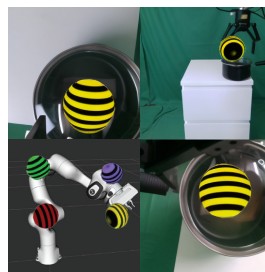

*Figure 13.* Tiled prediction with virtual robot-state (bottom left).

**Slow speed of diffusion agent.** The diffusion agent runs at a considerably lower frequency (5 Hz) than the controller (50 Hz). This makes GENIMA less reactive and prone to error accumulation. But diffusion speed is unlikely to be a major issue in the future with rapid-advances from the image-generation community. Recent works like StreamDiffusion [100] run at 91.07 Hz on a single NVIDIA RTX 4090.

**Jerky motions.** The actions generated by GENIMA, especially right after generating a new target image, can sometimes result in jerky motions. This behavior is noticeable for some tasks in the supplementary videos. Such behavior could be a result of the agent not being trained enough. We also tried temporal ensembling [23] to smoothen outputs, but this hurt the ability to recover from mistakes. Future works could experiment with other smoothing techniques.

**Controller fails to follow targets.** Sometimes the controller visibly fails to reach the target provided by the diffusion agent. This could be because the controller does not know how to reach the target from the current state, given its limited training data. One solution could be to pre-train the controller to reach arbitrary robot-configurations to maximize the workspace coverage.

**Object rotations.** All RGB-to-joint agents in Section 3.1 struggle with tasks that randomize initial object poses with a wide-range of rotations. For instance, in `phone on base`, GENIMA achieves 19%, whereas 3D Diffuser Actor [40] achieves 94%. A small change to the phone's rotation, results in widely different trajectories for picking and placing it on the base. This effect could make behavior-cloning difficult. A potential solution could be to pre-train RGB-to-joint agents on tasks that involve heavy rotations such that they acquire some rotation-equivariant behavior.

**Discovering new behaviors.** Like all behavior-cloning agents, GENIMA only distills behaviors from human demonstrations, and does not discover new behaviors. It might be possible to fine-tune GENIMA on new tasks with Reinforcement-Learning (RL). Furthermore, advances in preference optimization [96, 101] for Stable Diffusion models could be incorporated to shape new behaviors.

**Hallucinations in predictions.** As with any image-generation framework, GENIMA's diffusion agent is prone to hallucinating visual artifacts. These hallucinations are limited to joint-action spheres since the background observations remain unchanged. A potential solution could be to use a classifier to detect out-of-distribution actions and then execute recovery behaviors.

---

[8]https://github.com/ros-visualization/rviz

## K  Things that did not work

In this section, we briefly describe things we tried but did not work in practice.

**Predicting target spheres at fixed intervals**. Instead of predicting spheres continuously at $t + K$ timesteps, we tried fixed intervals of $K$ e.g., 20, 40, 60 etc. These fixed intervals act as waypoints, where the spheres hover in place until the target is reached (instead of always being $K$ steps ahead). In this setting, controllers cannot be trained with random backgrounds, because the trajectory between fixed intervals is offloaded to the controller without any visual context. So we trained controllers with full context, but found that these controllers tend to ignore target spheres, and directly use the visual context for predicting actions.

**Other sphere shapes and textures.** We experimented with a few variations of target spheres. Our goal was to design simple shapes that are easy to draw with image-generation models, without having to draw the full robot with proper links and joints. Figure 14 illustrates an example in which we made the visual appearance more asymmetric. The sphere's surface is divided into octants with different colors and black dots. The dots indicate gripper open and close actions. But in practice, we found that a simple sphere with horizontal stripes works the best. Future works could improve GENIMA's performance by simply iterating on the target's appearance.

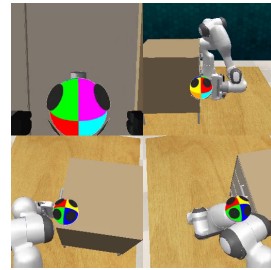

*Figure 14.* Sphere Octants.

**Controller with discrete actions.** We tried training controllers that output discrete joint actions similar to RT-2's [56] discrete end-effector actions. We discretized joint actions into bins. Each joint has its own prediction head and is trained with cross entropy loss. We found that the discrete controllers tend to generate smoother trajectories but lacks lack the precision for fine-grained tasks like manipulating small door handles or pressing tiny buttons.

**Observations with rendered current joint-states.** Instead of just RGB observations as input to the diffusion agent, we experimented with rendering the current joint-states with spheres as a visual reference. This led to agents with significantly worse performance, likely due to the input spheres occluding important visual information in the scene.

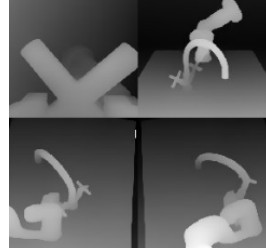

**Segmenting target spheres for the controller.** We experimented with providing binary segmentation masks of the target spheres to the controller. They had a negligible effect on success. Training with random backgrounds seems sufficient to make the controller focus on targets.

*Figure 15.* Depth by DepthAnything [102] from RGB input.

**Depth-based ControlNet.** We tried conditioning ControlNet on depth input instead of RGB. We used DepthAnything [102] to generate depth maps from RGB observations as shown in Figure 15. The performance with depth was worse or comparable to RGB input. Future works could experiment with fusing RGB and depth-based ControlNets [8].

## L  Safety Considerations

Real-robot systems controlled with Stable Diffusion [1] requires thorough and extensive safety evaluations. Internet pre-trained models exhibit harmful biases [103, 104], which may affect models fine-tuned for action prediction. This issue is not particular to Stable Diffusion, and even commonly used ImageNet-pretrained[9] ResNets [47] exhibit similar biases. Potential solutions include safety guidance [105] and detecting out-of-distribution or inappropriate generations with classifiers to pause action prediction and ask for human assistance. Keeping humans-in-the-loop with live visualizations of action predictions, and incorporating language-driven feedback, could further help in mitigating issues.

---

[9] https://excavating.ai/

