# OpenReview forum: "Generative Image as Action Models"
_robot-learning.org/CoRL/2024/Conference — CoRL 2024_

### Official Review · Reviewer_oFJa · 2024-07-21
**Using an image generator to predict future joint states for a robot to perform a task conditioned on text.**

**Originality:** 2
**Technical Quality:** 3
**Clarity Of Presentation:** 4
**Potential Impact:** 3
**Recommendation:** 3
**Confidence:** 3

**Review:**

Overall, the paper is clearly written, presents a decently novel approach, and sufficient evaluation.  There are multiple recent approaches to manipulation using image generators as priors. The paper compares against what it states as the most similar method, SuSIE, which generates images of the robot as sub-goals in contrast to GENIMA which predicts images of proxies for joint states. Based on their evaluations, it appears that it could be a better approach to generate these joint states rather than depictions of the robot.

Strengths:
The paper is fairly well organized and written.
It is an interesting idea to predict images of the joint states rather than the RGB appearance of the robot.
The evaluation is thorough. They compared against a similar method (SuSIE) and a state-of-the-art method (3D Diffuser Actor).
The experiment section poses interesting questions.

Weaknesses:
Many of the figures can be challenging to read based on font size. Figure 5 is particularly challenging.
The method does not outperform all methods. Though, the presented method requires less information that the superior methods which require more equipment to operate the robot.
perturbations_tests.mp4 is difficult to understand.

Typo Line 161: "outpeforms"
The title may read better as "Generative Images as Action Models"

**Quality Of The Limitations Section:**

3

**Questions For Rebuttal:**

In the 3D Diffusers Actor paper, SuSIE and 3D Diffuser Actor perform relatively similarly (Fig. 1 and Table IV). SuSIE is definitely inferior but the results are close. In your experiments, SuSIE and 3D Diffuser Actor are not even remotely close in performance. Could you please explain why?

You mention that 3D Diffuser Actor is not comparable exactly to your approach because it uses extra priors of depth, keypoints, scene bounds, and motion-planners. Are these pieces of information that you could extract with your current real-world setup? Do you think that if you were to incorporate these extra priors into your method, it could perform better or worse?

**Robotics Focus:**

4

**Summary Of Paper:**

This paper introduces a method to enable a robot to perform tasks conditioned on text. Their approach uses a pre-trained image generator. Rather than training the image generator to generate future goal states as images of the robot, it generates images of the joint states.

**Summary Of Recommendation:**

The paper introduces a novelty in a recent trend of using image generators as robot planners. I recommend weak accept due to the clarity of the paper.

---

### Official Review · Reviewer_u2x6 · 2024-07-22
**Important problem, interesting method, unclear motivation of method**

**Originality:** 4
**Technical Quality:** 4
**Clarity Of Presentation:** 4
**Potential Impact:** 3
**Recommendation:** 3
**Confidence:** 5

**Review:**

Strengths:
- The paper is well presented. The figures are well-made, and the presentation of the method, experimental setup, training data, etc., are very clear. I had no trouble understanding the idea with different level of details.

- The proposed method is innovative. The core of the idea is essentially parameterizing a robot's action as an image, which can be learned by a text-to-image generative model. It might not achieve SOTA results on hardcore manipulation tasks but it is an innovative attempt and the results seem promising.

- The use of SD-Turbo is neat. Being able to achieve 5 hz inference frequency is quite impressive. This could allow lots of potentially more impressive tasks to be done by the proposed method.

- The discussions of limitations are very thorough.

Weaknesses:
- Parameterizing robot actions as images have many limitations by construction. For example, different spheres can occlude each other, which leads to ambiguity in robot joint actions.

- Many prior works use large-scale internet-pretrained for image-editing or 3D tasks because of the impressive image priors or geometry priors learned by these large models. A similarity across all these tasks is that the output of the diffusion model is natural images so the task can benefit from the priors learned by the pretrained diffusion model. In this case, however, the images, made of four textured spheres with random background images, are not natural images. Therefore, it's hard to understand why pretrained diffusion models are needed. It's very likely a trained-from-scratch stable diffusion can perform equally well. It's also likely that a image generative model with much smaller architecture and less number of parameters can perform equally well but much more efficiently, allowing faster operation time. In summary, the reason to fine-tune from a pretrained large-scale diffusion model like stable diffusion is unclear here.

- Image generative models tend to hallucinate physically incorrect images. How does the method cope with strange output by the Genima model? Can the learned policy recover itself from incorrect prediction in a closed-loop setting? Are there example videos showing the recovery behavior?

**Quality Of The Limitations Section:**

3

**Questions For Rebuttal:**

- How will the results look like if the diffusion model is trained from scratch with the same data? Same goes to if the generative model is a much smaller one.

- Why do you need 4 textured spheres to represent the robot body? For most of the tasks, only the end effector poses are relevant to its success. Given the 4 colored spheres, how do you resolve the inconsistency between the generated colored spheres?

**Robotics Focus:**

4

**Summary Of Paper:**

The paper proposes to finetune text to image diffusion model as a visuomotor policy for planning robot actions during task execution. Specifically, the proposed method parameterized robot actions as four patterned spheres representing the 6 dof locations, orientations, and states of the robot joints and use a dataset to finetune a stable diffusion model with controlnet. Real worlds experiments were conducted to validate the effectiveness of the proposed method.

**Summary Of Recommendation:**

I like the idea and the method of the paper and I think it could potentially work better in the future if the limitations are addressed. However, the justification of using large-scale pretrained image diffusion model is not well-established. I would believe a smaller model with less weights can achieve similar results with higher inference frequency thus potentially be able to handle dynamic tasks.

---

### Official Review · Reviewer_LDdg · 2024-07-29
**Interesting Idea**

**Originality:** 4
**Technical Quality:** 4
**Clarity Of Presentation:** 4
**Potential Impact:** 3
**Recommendation:** 3
**Confidence:** 3

**Review:**

Strength:
- The idea of using diffusion model to draw joints on images is an interesting and novel
- The proposed method offers better interpretability compared to other methods
- Sufficient experiments in both sim and real, demonstrating that the proposed method outperforms the baseline methods.

Weaknesses:
- I think the authors may consider including a metric for the accuracy of the predicted sphere poses. Although it might be tricky as the generation is in image space and poses are not explicitly represented (maybe through a pose estimator, or a differentiable renderer). This metric could help identify the performance bottleneck, whether it stems from errors in the image generator or the controller, and facilitate better diagnosis when the model fails. It could also be used to evaluate whether the generated joint poses are feasible and consistent across different views (see below).
- It would be good to include an additional ablation experiment with an open-loop trajectory follower, where the robot directly follows the estimated joints (sphere) poses, if these poses are available.
- The method may fail if parts of the joints are out of the field of views, and it is unclear if the generated configurations are consistent across different views.

**Quality Of The Limitations Section:**

3

**Questions For Rebuttal:**

Questions:
- I am curious why the authors chose striped spheres instead of other shapes, such as keypoints or axis.
- Regarding the stripes on the spheres, is the northern hemisphere asymmetric to the southern hemisphere?
- What is the direct output of the diffusion model (the output of the VAE decoder)? Is it only the spheres, or are the spheres blended with the background image of the robot? If it is the latter, how is it ensured that the background part matches the input image?

**Robotics Focus:**

4

**Summary Of Paper:**

This paper proposes a novel approach using diffusion-based image generation for visuomotor control. The authors propose to fine-tune a pre-trained diffusion model with ControlNet to draw target joints (represented as striped spheres) onto observed RGB images. These synthesized joint images are then used to generate actions through an ACT controller. Experiments conducted in both simulation and real-world demonstrate the effectiveness of the proposed method compared to baseline methods.

**Summary Of Recommendation:**

Overall, I think this is an interesting paper, and the experimental results demonstrate the effectiveness of the proposed method. Therefore, I recommend accepting this paper.

---

### Author Rebuttal · Authors · 2024-08-08

We thank the reviewers for their insightful comments and positive feedback. All reviewers highlighted that the paper is “novel”, “interesting”, and “well organized”.  Reviewer LDdg and oFJa noted that the evaluations were thorough. Reviewer u2x6 highlighted that GENIMA could “ allow lots of potentially more impressive tasks to be done”.

Here we address one common concern:

**Occluded and out-of-view sphere targets.**
While occlusion and out-of-view sphere targets are a concern, they are not common. With four cameras, each joint is visible at least from some viewpoint. The joints are far apart enough that severe occlusion is rare. Please checkout our website ([https://genima-bot.github.io/](https://genima-bot.github.io/)) for rollouts to get a sense of these generated spheres. One potential solution is to render virtual robot-states as shown in Figure 13 of the appendix.

Please see the comments below for individual responses.

---

### Decision · Program_Chairs · 2024-09-04

**Decision:**

Accept

**Comment:**

Strengths:
- Clear presentation;
- Interesting and novel idea.

Weaknesses:
- Potential limitations inherent to the construction of the method;
- Additional comparisons and ablations may further strengthen the experiment section.

Post rebuttal:
All three reviewers recommended acceptance, despite some valid concerns from reviewer u2x6.  Weighing all the pros and cons, the AC recommends acceptance, but encourages the authors to acknowledge the limitations in the camera ready as pointed out by reviewers.